# META GRADIENT BOOSTING NEURAL NETWORKS

## ABSTRACT

Meta-optimization is an effective approach that learns a shared set of parameters across tasks for parameter initialization in meta-learning. A key challenge for meta-optimization based approaches is to determine whether an initialization condition can be generalized to tasks with diverse distributions to accelerate learning. To address this issue, we design a meta-gradient boosting framework that uses a base learner to learn shared information across tasks and a series of gradient-boosted modules to capture task-specific information to fit diverse distributions. We evaluate the proposed model on both regression and classification tasks with multi-mode distributions. The results demonstrate both the effectiveness of our model in modulating task-specific meta-learned priors and its advantages on multi-mode distributions.

## 1 INTRODUCTION

While humans can learn quickly with a few samples with prior knowledge and experiences, artificial intelligent algorithms face challenges in dealing with such situations. Learning to learn (or meta-learning) (Vilalta & Drissi, 2002) emerges as the common practice to address the challenge by leveraging transferable knowledge learned from previous tasks to improve learning on new tasks (Hospedales et al., 2020).

An important direction in meta-learning research is meta-optimization frameworks (Lee & Choi, 2018; Nichol & Schulman, 2018; Rusu et al., 2019), a.k.a., model-agnostic meta-learning (MAML) (Finn et al., 2017). Such frameworks learn initial model parameters from similar tasks and commit to achieving superior performance on new tasks that conform to the same distribution through fast adaptation. They offer excellent flexibility in model choices and demonstrate appealing performance in various domains, such as image classification (Li et al., 2017; Finn et al., 2017), language modeling (Vinyals et al., 2016), and reinforcement learning (Fernando et al., 2018; Jaderberg et al., 2019).

Generally, such frameworks define a target model $\mathcal{F}_\theta$ and a meta-learner $\mathcal{M}$. The learning tasks $\mathcal{T} = \{\mathcal{T}^{train}, \mathcal{T}^{test}\}$ are divided into training and testing tasks, where $\mathcal{T}$ are generated from the meta-dataset $\mathcal{D}$, i.e., $\mathcal{T} \sim P(\mathcal{D})$. Each task contains a support set $D^S$ and a query set $D^Q$ for training and evaluating a local model. The initialization of the model parameter $\theta$ is learned by the meta learner, i.e., $\theta \leftarrow \mathcal{M}(\mathcal{T}^{train})$. We denote the meta-learned parameter as $\phi$ so that $\theta \leftarrow \phi$. For each task, the model obtains locally optimal parameter $\hat{\theta}$ by minimizing the loss $\mathcal{L}(\mathcal{F}_\theta(D^S))$. The meta parameter $\phi$ will be updated across all training tasks by minimizing the loss $\Sigma_{T \in \mathcal{T}^{train}}(\mathcal{L}(\mathcal{F}_{\hat{\theta}}(D^Q)))$. Generally, it takes only a small number of epochs to learn locally optimal parameters across training tasks so that meta-learned parameter $\phi$ can quickly converge to an optimal parameter for new tasks.

Most methods assume some transferable knowledge across all tasks and rely on a single shared meta parameter. However, the success of the meta-learners are limited within similar task families, and the single shared meta parameter cannot well support fast learning on diverse tasks (e.g., a large meta-dataset) or task distributions (e.g., $\mathcal{T}$ are generated from multiple meta-datasets) due to conflicting gradients for those tasks (Hospedales et al., 2020). Recent efforts have studied multiple initial conditions to solve the above challenges. Some employ probabilistic models (Rusu et al., 2019; Finn et al., 2018; Yoon et al., 2018) while others incorporate task-specific information (Lee & Choi, 2018; Vuorio et al., 2019; Alet et al., 2018). The former learns to obtain an approximate posterior of an unseen task yet needs sufficient samples to get reliable data distributions; the latter conducts task-specific parameter initialization using multiple meta-learners yet requires expensive computation and cannot transfer knowledge across different modes of task distributions.

In this work, we aim to resolve the above challenges from a novel perspective by proposing a meta gradient boosting framework. Gradient boosting (Friedman, 2001) aims to build a new learner towards the residuals of the previous prediction result for each step. We call the learner for each step as *weak learner* and make predictions based on summing up the weak learners. Recent research (Badirli et al., 2020; Olson et al., 2018) has demonstrated the potential of decomposing deep neural nets into an ensemble of sub-networks with each achieving low training errors. We propose to use the first or first few weak learners as the base learner, followed by a series of gradient boosting modules to cope with a diverse array of tasks—the base learner is responsible for inferring transferable knowledge by learning across all tasks; the gradient-boosting modules are designed to make task-specific updates to the base learner. Compare with existing work, which uses multiple initial conditions, our approach does not require specifying a set of initialization conditions and thus has better flexibility in dealing with multi-mode tasks. Our proposed framework is also more efficient than its counterparts as it does not require a large number of gradient boosting modules. We evaluate the proposed framework on few-shot learning scenarios for both regression and classification tasks. The experimental results show the well performance of the proposed framework, which demonstrates the model's ability in learning with very few cases.

## 2 RELATED WORK

Meta-learning has the potential of replicating human ability to learn new concepts from one or very few instances. It has recently drawn increasing attention, given its broad applicability to different fields (Hospedales et al., 2020). Pioneers (Finn et al., 2017; Nichol & Schulman, 2018) in this topic propose optimization algorithms with learned parameters to automate the exploitation to the structures of learning problems. However, most of them initialize the same set of model parameters for all tasks, which may have different distributions, thus resulting in over-fitting.

Recent studies either model the mixture of multiple initial conditions via probabilistic modeling (Finn et al., 2018; Yoon et al., 2018) or incorporate task-specific knowledge (Lee & Choi, 2018; Alet et al., 2018), to address the above issues. Yoon et al. (2018) and Finn et al. (2018) use variational approximation to enable probabilistic extensions to MAML. But it is unclear how to extend MAML for a wide range of task distributions. Rusu et al. (2019) consider multiple conditions by borrowing the idea of variational autoencoders (Kingma & Welling, 2014), which encodes inputs to a low-dimensional latent embedding space and then decodes the learned latent code to generalize task-specific parameters. Another line of research defines a set of initialization modules and incorporate task-specific information to select task-specific modules; this way, it can identify the mode of tasks sampled from a multimodal task distribution and adapt quickly through gradient updates (Vuorio et al., 2019). Yao et al. (2019) propose a Hierarchically Structured Meta-Learning (HSML) framework to perform soft clustering on tasks. HSML first learns the inputs and then obtains clustering results by the hierarchical clustering structure. HSML tailors the globally shared parameter initialization for each cluster via a parameter gate to initialize all tasks within the clusters. The above approaches have common limitations in 1) requiring sufficient data samples to generalize task distribution thus may fail in few-shot cases; 2) being computationally expensive, due to the globally stored initialization modules; 3) facing challenges in exhaustively listing every possible initial condition.

Two closely-related topics to meta-learning are modular approaches (Andreas et al., 2016) and multi-task learning (Zhang & Yang, 2017). Modular approaches are similar to meta-learning in that the input signal gives relatively direct information about a good structural decomposition of the problem. For example, Alet et al. (2018) adopt the modular structure and parameter adaptation method for learning reusable modules. Multi-task learning aims to learn a good shared-parameter or make the parameter for each task as similar as possible (Wang et al., 2020). For example, Zhang et al. (2018) propose two task networks that share the first few layers for the generic information before applying different prediction layers to different tasks. These approaches differ from meta-learning in requiring fine-tuning the models over all training samples and thus cannot adapt well to new tasks.

Our framework Meta Gradient Booting (MGB) neural network is based on the idea of gradient boosting (Friedman, 2001), which aims to build a new learner towards the residuals of the previous prediction result for each step. The learner for each step is called weak learner, and the prediction is based on the summation of weak learners. Weak learners may vary from traditional decision trees (Chen & Guestrin, 2016) to neural networks (Tannor & Rokach, 2019; Badirli et al., 2020).

**Algorithm 1** Training of MGB
1: Randomly initialize global parameter $\phi$
2: **while** Not done **do**
3:     **for** $T \in \mathcal{T}$ **do**
4:         **for** $(x, y) \in D^S$ **do**
5:             Initialize $f_{\theta_0}$ by $\theta_0 \leftarrow \phi$
6:             **for** $k \in range(K)$ **do**
7:                 $\theta \leftarrow \theta - \beta \mathcal{L}(y, \mathcal{F}_\theta)$
8:             **end for**
9:         **end for**
10:         Get updated parameter $\hat{\theta}$
11:         **for** $(x, y) \in D^Q$ **do**
12:             Calculate predictions $\mathcal{F}_{\hat{\theta}}(x)$
13:             Calculate task loss $\mathcal{L}(y, \mathcal{F}_{\hat{\theta}})$
14:         **end for**
15:     **end for**
16:     Update $\phi$ by $\phi \leftarrow \phi - \gamma \mathcal{L}_{meta}$
17: **end while**

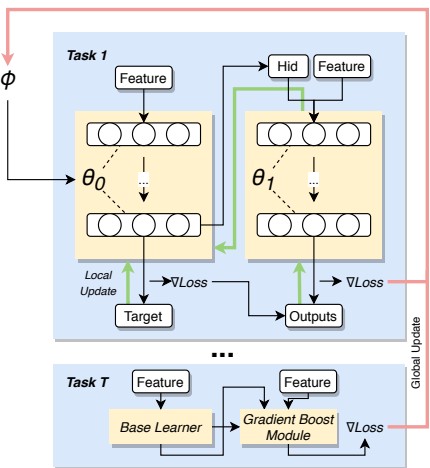

Figure 1: Example of the model with only one gradient-boosting module. Green lines are for local update and red lines are for global update.

A recent study (Badirli et al., 2020) proposes a general framework for gradient boosting on neural networks, which work for both regression and classification tasks. It uses the deep layers of neural nets as a bagging mechanism in a similar spirit to random forest classifier (Veit et al., 2016). After only slight tuning, deep neural nets can perform well on a wide range of small real-world datasets (Olson et al., 2018). These findings demonstrate the potential of decomposing deep neural nets into an ensemble of sub-networks each achieving low training errors. In our framework, we use the first weak learner or the first few weak learners as the base learner for learning the shared initialization parameter across tasks. The output for each weak learner is then aggregated to the inputs for the next step for constructing an end-to-end learning strategy until the last gradient boosting module. This way, the base learner serves as transferable knowledge, and the gradient boosting modules following it are trained for task-specific predictions.

## 3 METHOD

We explore the problem in the context of supervised learning, where input-output pairs are available in both training and validation sets. Similar to previous meta-optimization based approaches (Finn et al., 2017; Nichol & Schulman, 2018), we assume the tasks are generated from an underlying distribution $\mathcal{T} \sim P(\mathcal{D})$, where $\mathcal{D}$ is the meta-dataset, which is either a uni-mode dataset or multi-mode datasets. Given a set of tasks $\mathcal{T} = \{\mathcal{T}^{train}, \mathcal{T}^{test}\}$, each task $T \in \mathcal{T}$ contains a support dataset $D^S$ and a query dataset $D^Q$, both representing input-output pairs $(x, y)$. We aim to learn a meta-learner $\mathcal{M}$ to guide the initialization for a target model $\mathcal{F}_\theta$ so that the target model can quickly adapt and perform well on a given new task. We propose a Meta Gradient Boosting (MGB) framework as the target model $\mathcal{F}_\theta$, which consists of several weak learners and can be represented as $\mathcal{F}_\theta \sim \Sigma_{k=0}^K f_{\theta_k}$. The first weak learner $f_{\theta_0}$ or the first few weak learners are regarded as the base learner for learning the shared information across tasks; the weak learners are gradient boosting modules for capturing task-specific information. The meta learner aims to learn transferable knowledge and provides initialization details for the base learner so that the model can quickly adapt to task-specific predictions with a few gradient-boosting steps. Figure 1 shows an example of our MGB framework under $K = 1$, where we update the model locally for task-specific predictions and update the meta-learner globally for all tasks.

### 3.1 LOCAL LEARNING: TASK-ADAPTIVE UPDATING VIA GRADIENT-BOOSTING

Gradient boosting machines hold out optimization in the function space (Friedman, 2001). In particular, the target model is an addition $\mathcal{F}_\theta = \Sigma_{k=0}^K \alpha_k f_{\theta_k}$, where $K$ is the number of adaptions

(gradient boosts), $f_{\theta_0}$ is the first weak learner, which provides initial prediction of the inputs, $f_{\theta_k}$ are the function increments (gradient boosting modules), and $\alpha_k$ is the boosting rate. In each step, the new gradient boosting module is formulated in a greedy way. To start, the base-learner $f_{\theta_0}$ minimizes a prediction loss $\mathcal{L}(y, f_{\theta_0})$ to predict the outputs $\hat{y} \leftarrow f_{\theta_0}(x)$. Then, at gradient boosting step $k$, the gradient boost module $f_{\theta_k}$ minimizes the loss $\mathcal{L}(g_k, f_{\theta_k})$, where $g_k = -\frac{\partial \mathcal{L}(y, \mathcal{F}_\theta^{k-1}(x))}{\partial \mathcal{F}_\theta^{k-1}(x)}$, $\mathcal{F}_\theta^k = \Sigma_{*=0}^k \alpha_* f_{\theta_*}$ denotes the ensemble of functions at gradient step $k$, and $g_k$ is the negative gradient along with the observed data. Traditional boosting frameworks learn each weak learner greedily; therefore, only parameters of $k$-th weak learner are updated at boosting step $k$ while all the parameters of previous $k-1$ weak learners remain unchanged. This together with the single shared meta parameter make it easy for the model to stuck in a local minimum. The fixed boosting rate $\alpha_k$ further aggravates the issue. In response, we construct the gradient boosting neural network in a cascading structure (He et al., 2016). Similar to Badirli et al. (2020), at each step $k$, we take the concatenation of the inputs $x$ and the hidden layer of the previous weak learner $h_{k-1} = \sigma_{\theta_{k-1}}(x)$ as the inputs for the current step gradient boost module, i.e., $g_k \leftarrow f_{\theta_k}([h_{k-1}, x])$. But our approach differs in optimizing the gradient boosting neural networks in a macroscopic view—for each step $k$, we learn the ensemble of the weak learners $\mathcal{F}_\theta^k$ by minimizing the loss function

$$\arg\min_\theta \mathcal{L}(y, \mathcal{F}_\theta^k) \rightarrow \arg\min_\theta \mathcal{L}(y, \alpha_0 f_{\theta_0}(x) + \Sigma_{k=1}^K \alpha_k f_{\theta_k}(h_{k-1}, x)), \tag{1}$$

We update parameters of both weak learners and gradient boost module via back-propagation. Generally, the parameter $\theta$ of $\mathcal{F}_\theta$ is locally updated via $\theta \leftarrow \theta - \beta \mathcal{L}(y, \mathcal{F}_\theta)$, where $\beta$ is the task learning rate. The boosting rate $\alpha_k$ can be set in various forms—in the simplest case, we can use an increasing or decreasing boosting rate, e.g. $\alpha_k = \alpha_{k-1}/c$ ($c$ is a constant), to decrease or increase the contribution of the base learner. We will discuss model performance under different settings of the boosting rate later. Both the boosting rate and the number of gradient boost modules affect the sharing ability and prediction performance of the base learner. Hochreiter et al. (2001) found that the gradient for the earlier weak learners decays with the increasing number of gradient boost modules. On balance, we use the base learner of our proposed gradient boosting framework as a weak learner and a series of gradient boosting modules as strong learners for a specific task.

## 3.2 GLOBAL LEARNING: META-OPTIMIZATION FOR TRANSFERABLE KNOWLEDGE LEARNING

The learning and updating strategy of the gradient boosting framework ensure a weak base learner. Since the base learner could be the first weak learner or the first few weak learners, we use $f_{\theta_0}$ to represent for both conditions for ease of illustration. We take the meta-optimization approach for initializing the base learner so that the model can provide an initial guess for the prediction based on the transferable knowledge over a wide arrange of tasks. Specifically, we learn a global sharing parameter $\phi$ s.t. $\theta_0 \leftarrow \phi$. Similar to other MAML-based approaches (Finn et al., 2017; Lee & Choi, 2018), we minimize the expected loss on the local query set $D^Q$ for tasks $T \in \mathcal{T}^{train}$ in meta optimization. Since the meta-gradient may involve higher-order derivatives, which are computationally expensive for deep neural nets, MAML Finn et al. (2017) takes one-step gradient descent for meta-optimization. Following the above, we obtain updated model parameters $\hat{\theta}$ after updating the target model $\mathcal{F}_\theta$ for $K$ steps on the local support set $D^S$. Global learning aims at minimizing the loss

$$\arg\min_\phi \mathcal{L}_{meta} \rightarrow \arg\min_\phi \Sigma_{T \in \mathcal{T}^{train}} \Sigma_{(x,y) \in D^Q} \mathcal{L}(y, \mathcal{F}_{\hat{\theta}}) \tag{2}$$

The global sharing parameter $\phi$ is updated via $\phi \leftarrow \phi - \gamma \mathcal{L}_{meta}$, where $\gamma$ is the global learning rate. The pseudocode of the training process is described in Algorithm 1.

## 4 EXPERIMENTS

We test our proposed framework on few-shot learning scenarios and compare it with three other meta-learning approaches: MAML (Finn et al., 2017), Multimodal Model-Agnostic Meta-Learning (MMAML) (Vuorio et al., 2019), and Meta-learning with Latent Embedding Optimization (LEO) Rusu et al. (2019). Both MMAML and LEO model a mixture of multiple initial conditions. MMAML modulates its meta-learned prior parameters according to the identified mode to enable more efficient

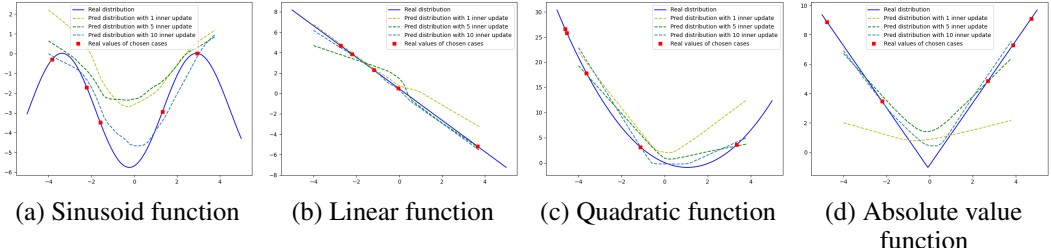

(a) Sinusoid function     (b) Linear function     (c) Quadratic function     (d) Absolute value function

Figure 2: Performance on 4-mode regression tasks with one step of global learning. Blue lines show the real distribution. Red nodes stand for the real values of the support set samples.

adaptation; LEO solves the problem via probabilistic modeling that learns a stochastic latent space with an information bottleneck, conditioned on the input data, from which the high-dimensional parameters are generated. We compare the results on both regression and classification tasks.

## 4.1 REGRESSION TASKS

**Setups** We adopt the simple regression task with similar settings to Finn et al. (2018). In the 1D regression problem, we assume different tasks correspond to different underlying functions, and we aim to predict the outputs for new tasks by learning on very few samples. To construct a multi-mode task distribution, we set up four functions (sinusoidal, linear, quadratic, and abstract value functions) and treat them as discrete task modes. The function settings are detailed in Appendix A.1. Uni-mode training tasks are generated from a single function. For each task, input-output pairs are generated from the given function with randomly initialized parameters (e.g., the slope for the linear function). We generate multi-mode training tasks by first selecting the function type for each task and then initializing random parameters for the function to generate samples. Similar to the settings in Finn et al. (2017), we use two hidden layers of size 40, followed by ReLU as the activation function.

Learning from multi-mode distributions (e.g., the above four distributions) is challenging. First, the function values may vary across those distributions. For example, the output of the quadratic function can range from zero to dozens (see Figure 2) while the other three functions are more likely to produce outputs within [-10,10]. Second, the few-shot samples that sit on a line might be generated from non-linear functions, which make it difficult to learn the real modality from such samples. Updating more steps for task-specific models could solve the first challenge yet may cause over-fitting. The second challenge can be mitigated by producing a set of models for task learning. Our proposed MGB can well handle the two challenges by a series of task-specific gradient boost modules and providing flexibility in updating the framework.

**Results** We use Mean Absolute Error (MAE) as the evaluation metric to evaluate models' performance in training on one-mode (sinusoidal function), two-mode (sinusoidal function and linear function) or four-mode (all the four functions) tasks. To ensure a fair comparison, the same basic settings of the network structure are configured for all the compared methods, and the results are learned within certain training epochs. Detailed settings are in A.3. Overall, the proposed MGB framework shows competitive and stable performance on multi-mode regression tasks. The results (shown in Table 1) reveal that it is more difficult to capture task-specific patterns on multi-mode tasks—the MAE is larger when more modes are considered. This makes sense considering the randomness in choosing functions and function parameters for all models. The results also show that incorporating task identities can significantly improve the performance of multi-mode learning. MAML has the highest error in all settings, which suggests that MAML does not perform as well on multi-mode tasks as on uni-mode tasks. Our model shows more stable performance in all settings when compared with LEO and MMAML. Performance comparison of our model with one, two, and five gradient boosting modules (corresponding to MGB-1, MGB-2, and MGB-5, respectively) suggests that the performance improves when more gradient boosting modules are used; however, the effect decreases as the number gradient boosting modules increases.

Table 1: Results on Regression Tasks (MAE)

| Method | 1 Mode | | 2 Modes | | 4 Modes | |
|---|---|---|---|---|---|---|
| | 5-shot | 10-shot | 5-shot | 10-shot | 5-shot | 10-shot |
| MAML | 1.234±0.174 | 1.054±0.077 | 1.548±0.223 | 1.356±0.109 | 2.044±0.472 | 1.698±0.267 |
| LEO | 0.957±0.123 | 0.789±0.042 | 1.127±0.175 | 0.899±0.084 | 1.234±0.248 | 1.095±0.163 |
| MMAML | 0.638±0.053 | 0.526±0.027 | 0.783±0.096 | 0.709±0.048 | 1.016±0.181 | 0.920±0.099 |
| MGB-1 | 0.674±0.009 | 0.524±0.013 | 0.999±0.103 | 0.906±0.038 | 1.213±0.173 | 0.928±0.084 |
| MGB-2 | 0.629±0.004 | 0.512±0.005 | 0.822±0.032 | 0.734±0.021 | 1.046±0.106 | 0.899±0.027 |
| MGB-5 | **0.615±0.005** | **0.476±0.005** | **0.704±0.042** | **0.672±0.077** | **0.985±0.089** | **0.825±0.043** |

Table 2: Results on Classification Tasks (Accuracy)

| Method | 2 Modes | | | 3 Modes | | | 4 Modes | | |
|---|---|---|---|---|---|---|---|---|---|
| | 5-way | | 20-way | 5-way | | 20-way | 5-way | | 20-way |
| | 1-shot | 5-shot | 1-shot | 1-shot | 5-shot | 1-shot | 1-shot | 5-shot | 1-shot |
| MAML | 0.6381 | 0.7524 | 0.4296 | 0.5235 | 0.6481 | 0.2986 | 0.4223 | 0.5172 | 0.2415 |
| LEO | 0.6676 | 0.7689 | 0.4318 | 0.5129 | 0.6413 | 0.3106 | 0.3948 | 0.4757 | 0.2301 |
| MMAML | 0.6797 | 0.7738 | **0.4521** | 0.5536 | 0.6728 | 0.3534 | **0.4812** | 0.5528 | **0.3149** |
| MGB-1 | 0.6394 | 0.7579 | 0.4228 | 0.5241 | 0.6435 | 0.3002 | 0.4277 | 0.5209 | 0.2533 |
| MGB-2 | 0.6501 | 0.7633 | 0.4270 | 0.5503 | 0.6750 | 0.3459 | 0.4531 | 0.5374 | 0.2682 |
| MGB-5 | **0.6834** | **0.7830** | 0.4426 | **0.5611** | **0.6897** | **0.3568** | 0.4725 | **0.5530** | 0.2956 |

## 4.2 CLASSIFICATION TASKS

**Setups** We evaluate the proposed framework (Finn et al., 2017) on $n$-way few-shot image classification tasks. We use four datasets to constitute multi-mode tasks: Omniglot Lake et al. (2011), miniImageNet Ravi & Larochelle (2016), FC100 Oreshkin et al. (2018), and CUB Wah et al. (2011). Details about those datasets can be found in the supplementary material A.2. We follow train-test splitting methods as described in Finn et al. (2017); Vuorio et al. (2019) and train models on tasks with two modes (Omniglot and miniImageNet), three modes (Omniglot, miniImageNet, and FC100), and four modes (all four datasets) tasks. The weak-learner uses CNN modules for learning the image embedding and fully-connected layers for classification. Similar to previous work Finn et al. (2017); Vuorio et al. (2019), we configure each CNN module with $3 \times 3$ convolutions, followed by $2 \times 2$ max-pooling and batch normalization (Ioffe & Szegedy, 2015). Since the embedding module can significantly affect classification results (Sun et al., 2019), to ensure a fair comparison, we use the same embedding module for all compared methods. The detailed settings are described in A.3.

**Results** We consider 1-shot and 5-shot learning for 5-way classification and 1-shot learning for 20-way classification. We evaluate the performance of our MGB framework with one (MGB-1), two (MGB-2), or five (MGB-5) gradient boosting modules. The resulting performance is shown in table 2. Overall, the proposed MGB performs well on multi-mode tasks. Compared with MMAML, our method works better on most scenarios except on 1-shot 20-way classifications, where MMAML can store more parameters. Similar to the regression tasks, MGB with more gradient boosting modules shows better performance while MGB-1 can make only a slight improvement over MAML because images contain more complex information than real numbers. More modes of tasks increase the performance gap between MAML and the other methods, which suggest the other methods (which consider multiple conditions) can handle multi-mode tasks better than MAML. Under the same experimental settings, i.e., with the same image embedding modules, LEO does not perform well on tasks with more modes, partially because it is largely impacted by the quality of the learned image embedding—first, LEO's learning strategy (Rusu et al., 2019) pretrains the dataset-specific image embedding (Qiao et al., 2018) before meta-learning; then, LEO uses an encoder-decoder structure to generate parameters for the classifier from the learned embedding.

## 5    DISCUSSION

Our experimental results on both regression and classification tasks suggest our method can adapt to the optimal results with few gradient boosting modules. In this section, we take a further step to discuss i) the configuration of gradient boosting modules and ii) the sharing ability of the base learner during the back-propagation through meta-gradient boosting modules. The results are presented for 5-way 1-shot image classification with 4-mode tasks.

### 5.1    CONFIGURATION OF GRADIENT BOOSTING MODULES

**Settings for the weak learner**    Our MGB framework consists of a series of weak learners, where the first or the first few weak learners serve as the base learner to be shared among tasks. Type and dimension of the weak learner are two key factors that may affect the final results. Vuorio et al. (2019) find that LSTM models perform better than linear models in regression tasks. Sun et al. (2019) show the choice of feature extractor for images has a strong correlation with the final results, and using pre-trained network or network structure can improve the results significantly. For example, it improves the accuracy by about 6% (Mishra et al., 2018; Oreshkin et al., 2018) to use ResNet-12 as the feature extractor. Here, we use four convolutional layers for learning from images to ensure a fair comparison. The dimension of the weak learner includes the number of hidden layers and the number of neurons in each layer. Figure 3 (a) shows the performance of the model trained on a 4-modes classification dataset under different settings of the two parameters above. The results show a deeper model or a larger neuron size gives better performance, while the network may need more time to learn with a larger neuron size.

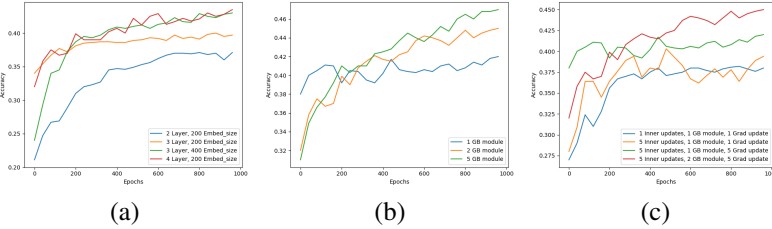

|  (a) | (b) | (c) |

Figure 3: Model performance under different settings on (a) the dimension of the weak learner, (b) the number of gradient boosting modules, and (c) the updating strategy for gradient boosting modules on 4-modes image classification tasks.

**Settings for gradient boosting modules**    The number of gradient boosting modules and the updating strategy for the gradient boosting modules are two vital factors that may affect the final results. Since the prediction is based on the summation of a series of weak learners, more gradient boosting modules will reduce the contribution of each weak learner. According to the results shown in Figure 3 (b), the model with only one gradient boost module cannot well handle multi-mode tasks; more gradient boost modules can help improve the results while taking more time to learn the task information. The traditional gradient boost module greedily updates the whole framework at each step, but we can let the added weak learner grasp more information from the inputs by updating several times in each step. We can further adjust the contribution of the base learner and gradient boosting modules to the final results by allowing them to update for different times in each step. Intuitively, if we fine-tune on the base learner (i.e., updating multiple times on the base learner), the model may get stuck in a local optimal and decrease the impact of gradient boosting modules on the model performance; conversely, if we conduct more updates on the gradient boosting modules, the model may need more training epochs from all training tasks to grasp the general sharing information that helps the model in fast-adaptation. The results under different settings of the updating times for base learner and gradient boosting modules are shown in Figure 3 (c). We can see the updating strategy significantly affects model performance, and updating more steps on gradient modules improves the model's robustness. The performance tends to become unstable if we conduct more updates for the base learner (i.e., the yellow line in Figure 3 (c)), which observation aligns with our analysis above.

## 5.2 SHARING ABILITY OF THE BASE LEARNER

The base learner of our MGB framework is shared across tasks for capturing the general sharing knowledge. Three factors may affect the sharing ability of the base learner: 1) which part to share; 2) how to share; 3) how the shared base learner contributes to MGB. We discuss these three components as follows.

**Single weak learner v.s. Multiple weak learners** Instead of choosing a single weak learner as the base learner, we can choose the first few weak learners as the base learner. We present the results of MGB with one, two, or three weak learners as the base learner and one gradient boosting module in Figure 4 (a). Generally, when more weak learners are used as the base learner (which is more than the number of gradient boost modules), MGB faces difficulties in capturing multi-mode patterns and thus achieves degraded generalization performance.

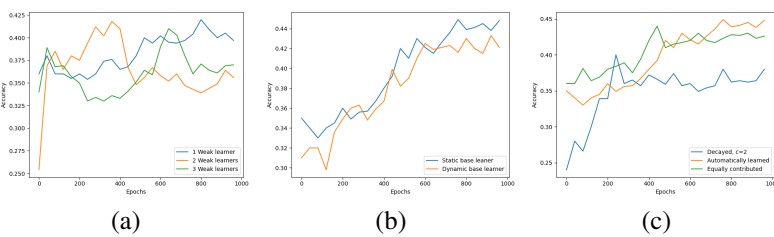

(a)             (b)             (c)

Figure 4: Model performance under different settings for the base learner with respect to (a) the number of weak learners, (b) the sharing strategy of the base learner, and (c) the choice for the boosting rate $\alpha$ on 4-modes image classification tasks.

**Static base learner v.s. Dynamic base learner** The base learner is initialized using the global sharing parameter $\phi$. It can be either static (if we keep its parameter unchanged) or dynamic (if we update the base learner during the training of gradient boost modules). We compare the versions of MGB that use a static base learner and a dynamic base learner, respectively. In both versions, we append one gradient boost module to the base learner and update it multiple times during each step. The results (shown in Figure 4 (b)) reveal that keeping the shared information (i.e. using static base learner) can improve the stability of the model.

**Boosting rate $\alpha$** The boosting rate ($\alpha$) is probably the most vital component for the MGB framework because it directly indicates the contribution of each weak learner to the final prediction. We test the performance of MGB under various settings of the boosting rate $\alpha$, where the rate is either decayed (i.e. $\alpha_k = \alpha_{k-1}/c$, where $c$ is a constant), automatically learned, or equally contributed (i.e. $\alpha_* = 0$ for all base learners), respectively. The result suggests that using the automatically learned $\alpha$ or equally contributed $\alpha$ leads to more stable performance, while a decayed $\alpha$ results in more time in task learning. This supports our analysis that our gradient boosting modules help learn task information.

## 6 CONCLUSION

In this work, we propose a novel direction for solving the problem faced by previous meta-optimization approaches, i.e., using the same initialization for diverse tasks. We present a meta gradient boosting framework that contains a series of weak learners to make predictions, using a base learner to grasp shared information across all tasks. We have conducted extensive experiments to evaluate the model's ability to capture meta information and the task-specific information on regression and classification tasks. Our experimental results show the effectiveness of our framework in learning multi-mode tasks. Our results reveal the necessity of selecting the weak learner carefully according to task types. An example is that CNNs outperform simple fully connected (FC) layers on image classification problems while FC layers can perform better on regression tasks. In future work, we will extend our framework by considering multi-modal problems, e.g., learning from images, texts and numerical values, and study how to choose appropriate weak learners for specific applications.

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

# A   IMPLEMENTATION DETAILS

## A.1   SETTINGS FOR REGRESSION TASKS

**Dataset details**   We use four functions for generating toy regression tasks: sinusoidal function, linear function, quadratic function, and absolute value function. For each task $T \in \mathcal{T}$, the selected function generates input-output pairs with randomly initialized parameters. The tasks $\mathcal{T}$ are generated from an underlying distribution $\mathcal{T} \sim P(\mathcal{D})$, where the meta-dataset $\mathcal{D}$ is the function set. Depending on the choice of the function set, we evaluate our model on uni-mode training (i.e. the tasks are generated from a single function with different random initialized parameters) and multi-mode training. To ensure each function can generate a variety of tasks, we define the four functions with three variables as follows:

- **Sinusoidal function:** $v_2 \cdot \sin(x + v_1) + v_3$
- **Linear function:** $v_2 \cdot (x + v_1) + v_3$
- **Quadratic function:** $v_1^2 \cdot x + v_2 \cdot x + v_3$
- **Absolute value function:** $v_2 \cdot |x + v_1| + v_3$

where $v_1$, $v_2$, and $v_3$ are sampled uniformly from ranges $[0, \pi]$, [-3,3], and [-3,3], respectively. The inputs are within [-5,5], and the outputs are perturbed with Gaussian noise with the standard deviation of 0.3. We sample 10 batches of 50 tasks for uni-mode and multi-mode training tasks, respectively.

**Model details**   Following Finn et al. (2017), we use two fully-connected layers of size 40 as the weak learner for regression tasks. The activation function is Leaky ReLU (Maas et al., 2013), which provides good and stable results over the four functions. By default, the MGB framework includes one weak learner as the base learner and three gradient boosting modules. We update ten times for both the base learner and gradient boosting modules. Both the task learning rate $\beta$ and the meta learning rate $\gamma$ are set to 0.01. We apply an automatic learning strategy to the boosting rate ($\alpha$), which is initialized to be 0.5. When constructing the gradient boosting neural network, we use a cascading structure (He et al., 2016) to make the network faster and stable. We use batch normalization(Ioffe & Szegedy, 2015) for the input for gradient boosting modules. Recall that at each step $k$, we take the concatenation of the inputs $x$ and the hidden layer of the previous weak learner $h_{k-1} = \sigma_{\theta_{k-1}}(x)$ as the inputs of gradient boost module at the current step, i.e., $g_k \leftarrow f_{\theta_k}([h_{k-1}, x])$. Specifically, we use shortcut layers to map the input $x$ to a latent layer $h_x$ with same shape as the hidden layer of the previous weak learner; then, we concatenate $h_x$ and $h_{k-1}$, which are both batch normalized, as the input for the current gradient boost module. We also test the impact of forwarding $[h_{k-1}, h_{k-2}]$ to gradient boost modules and find it only has a slight impact on model performance. In the default settings, we use $[h_{k-1}, x]$ as the input for each gradient boost module. Generally, it takes around one hundred epochs before getting acceptable results.

## A.2   SETTINGS FOR CLASSIFICATION TASKS

**Dataset details**   We use four datasets to evaluate multi-mode few-shot learning tasks: Omniglot (Lake et al., 2011), miniImageNet (Ravi & Larochelle, 2016), FC100 (Oreshkin et al., 2018), and CUB (Wah et al., 2011). Details about the datasets are listed in Table 3. We divide tasks in each dataset into train, validation or test tasks. Omniglot is a dataset of 1623 handwritten characters from 50 different alphabets, where we use the characters from the first 30 alphabets, the following 10 alphabets, and the last 10 alphabets as training, validating, and testing tasks, respectively. The miniImageNet dataset is a subset of the ImageNet[1]. Following Finn et al. (2017), we divide 100 classes into 64, 16, and 20 classes for meta-training, meta-validation, and meta-testing, respectively. Fewshot-CIFAR100 (FC100) is based on the widely applied CIFAR 100 dataset[2]. Similar to (Oreshkin et al., 2018), we divide 100 classes into 60 training classes, 20 validation classes, and 20 testing classes. As for the Caltech-UCSD Birds 200 dataset, i.e., the CUB dataset, we follow the settings in Vuorio et al. (2019) and use 140, 30 and 30 classes for training, validation, and testing, respectively. We generate uni-mode training tasks from one dataset. For each $n$-way $m$-shot learning task, we

---

[1] http://www.image-net.org/
[2] https://www.cs.toronto.edu/~kriz/cifar.html

Table 3: Basic information of the image datasets

| Dataset | Number of classes | Samples per class | Image channel | Image size |
|---|---|---|---|---|
| Omniglot | 1623 | 20 | 1 | 28×28 |
| miniImageNet | 100 | 600 | 3 | 84×84 |
| FC100 | 100 | 600 | 3 | 32×32 |
| CUB | 200 | >40 | 3 | 84×84 |

randomly draw $n$ classes from the dataset and $m$ samples from each class to form the support set. Then, we select from the unselected samples as the query set. We generate multi-mode training tasks by repeatedly run random selection of a dataset and drawing task samples for this dataset. In general, we use 10 batches of 50 tasks to train the model for both uni-mode tasks and multi-mode tasks. We resize the image for each dataset into 84×84 so that the model can share across the four datasets. Besides, we map images in Omniglot dataset to a layer with three channels so that they can be learned by the sharing model.

**Model details** The weak-learner uses CNN modules for learning the image embedding and fully-connected layers for classification. Similar to Finn et al. (2017), we configure each CNN module as 4 layers with $3 \times 3$ convolutions, followed by batch normalization (Ioffe & Szegedy, 2015), Leaky ReLU nonlinearity (Maas et al., 2013), and $2 \times 2$ max-pooling. The results of the CNN module are first mapped to a latent layer with 200 neurons and then predicted by a classifier with two fully-connected layers. The MGB framework, by default, includes one weak learner as the base learner and two gradient boosting modules. We update five times for both the base learner and the gradient boosting modules. The boosting rate ($\alpha$) for each gradient boost module is automatically learned and is initialized as 0.5. The task learning rate $\beta$ and the meta learning rate $\gamma$ are set to 0.001 and 0.005, respectively. Similar to the settings for regression tasks, we use shortcut layers for the input $x$ and a hidden layer of the previous weak learner as the input for the current gradient boost module. The model needs thousands of training epochs to obtain fair results. The results we presented for discussion are averaged for every tens of epochs.

## A.3 SETTINGS FOR COMPARED METHODS

We compare our work with three compatitve baselines: MAML (Finn et al., 2017), LEO (Rusu et al., 2019), and MMAML (Vuorio et al., 2019).

- MAML is one of the pioneer frameworks for meta optimization. For each task, the parameter $\theta$ of the task model $\mathcal{F}_\theta$ is initialized by the global sharing parameter $\phi$ and is updated by learning from the support set $D^S$. The meta parameter $\phi$ will be updated by learning from the loss on the query set $\mathcal{L}(D^Q)$.

- LEO use an external encoder-decoder to initialize the parameter $\theta$ of the task model $\mathcal{F}_\theta$. The external encoder-decoder framework has three parts: an encoder $g_{\phi_e}$, a relation network $g_{\phi_r}$, and a decoder $g_{\phi_d}$. For each task, the inputs are mapped to the latent code $z$ by $g_{\phi_e}$ and $g_{\phi_r}$; $\theta$ is initialized based on $g_{\phi_d}$ and the latent code $z$. When learning on a task, the latent code $z$ is first updated by the loss on support set (i.e., $z \leftarrow z - \beta\nabla_z\mathcal{L}(D^S)$) and then decoded to obtain the parameter $\theta$. Parameters of the encoder-decoder, i.e. $\phi_e$, $\phi_i$, and $\phi_r$, are updated globally by the query set loss on all tasks.

- MMAML first uses a modulation network to obtain the mode information of a sampled task and then uses this mode information to initialize the task model $\mathcal{F}_\theta$. The modulation network uses a task encoder $g_{\phi_h}$ to obtain a task embedding vector $v$ ($v \leftarrow g_{\phi_h}(x)$), which is then used to provide the modulation vectors $\tau$ for the parameters of the task model $\mathcal{F}_\theta$ (via $\tau^i = g^i_{\phi_g}(v)$). Suppose we have $I$ modules in the task model $\mathcal{F}_\theta$, and each module has parameter $\theta^i$, then each module will be updated as $\theta^i \leftarrow \theta^i \odot \tau^i$. For each task, the task model is first initialized by a meta parameter $\phi_m$, i.e., $\theta \leftarrow \phi_m$; then, each module of the task learner is adapted by $\theta^i \leftarrow \theta^i \odot \tau^i$, according to the outputs of the modulation network. The parameter $\theta$ is locally updated for each task, and the meta-parameters ($\phi_m$, $\phi_h$, and $\phi_g$) are updated across all tasks.

We choose the task learner $\mathcal{F}_\theta$ and configure the settings (including activation function, task learning rate $\beta$, and meta learning rate $\gamma$) for the compared methods in the same way as we configure our method on both regression and classification tasks. In particular, LEO uses pretrained embedding as input to statistically generate means and variances for the predictor in its encoder-decoder structure. The predictor can be regarded as one fully-connected layer that takes the pretrained embedding (Qiao et al., 2018) as input. The weights of the predictor are sampled from a distribution specified by the means and variances generated by LEO. Since LEO cannot be updated via classic back-propagation, we use similar measures as presented by Rusu et al. (2019). But we additionally use our strategy for generating the embeddings.

