# OpenReview forum: "Meta Gradient Boosting Neural Networks"
_ICLR.cc/2021/Conference — Reject_

### Official Review · AnonReviewer4 · 2020-10-23
**Marginal contribution**

**Rating:** 4
**Confidence:** 4

**Review:**

The authors propose a meta-gradient boosting framework that uses a base learner to learn shared information across tasks and gradient-boosted modules to capture task-specific information. The proposed approach is applied to various regression and classification tasks.

To me the motivation of using a gradient boosting framework remained unclear throughout the paper. It is stated in the last paragraph of the Introduction that ‘Recent research.. with achieving low training errors’. However, why this is important to the meta-learning problem remains unclear.


In Section 3, the model is not explained clearly:
-	What is a weak learner? In the introduction it is mentioned that ‘Gradient boosting aims to build a new learner towards…We call the learner for each step as weak learner.’ What is a step here?
-	‘..the first few learners are regarded as the base learner for learning the shared information across tasks..’ This was also not evident how it is designed to achieve this. Given that I have no prior on working with gradient boosting directly, I found the motivation and the method quite hard to follow.
-
Section 3.1. Second sentence: ‘..where K is the number of adaptions’-what is adaptions?
What is the definition of a gradient boost?
How is $\theta_k$ defined in equation (1)?

Sec 3.2:
The first sentence- what does it mean?
It is mentioned here that a first-order MAML type approach is taken to learn the ‘base learner’. But how the two goal of using gradient boost modules to ensure task-specific  information is achieved is not clear.

Figure 2,3,  and 4: The axes and fonts are unreadable.

In Results, the authors state that ‘ the results show that incorporating task identities can significantly improve the performance of multi-mode learning’. It was not clear to me how this is true. First, how is task identity even abstracted or assigned here? What features of the approach are able to get this information is not clear.

The gain classification tasks also seems marginal compared to the other approaches. The key emphasis of the authors is on the importance of the approach on multi-mode distributions. It is not made clear why the approach would be suited to this setting. This is supported numerically only in the case of regression experiments.

---

> ### Author Response · Authors · 2020-11-22
> **To reviewer 4**
>
> Regarding the **motivation and clarity of the gradient boosting model**, we suppose the audience to have some basic knowledge about the gradient boosting framework (Friedman, 2001). Besides, we have introduced the core idea of gradient boosting right at the start of the paper and given more details in our method. “Recent research …” suggests that the potential of decomposing deep neural nets into an ensemble of sub-networks; our framework of gradient boosting neural networks is one of such promising ensemble of sub-networks.
> 1.	Weak learner (defined in Section 3) and base learner (defined in Section 3.1) are two concepts in Gradient Boosting. The model makes predictions based on the weak learners, each of which can be considered as a neural network that provides its own prediction. Specially, the first or first few weak learners serve as the base learner to provide an initial guess for the prediction. We introduced the meaning of “step” in constructing a gradient boosting framework in lines 3-7 on page 4.
> 2.	The core idea of MAML is to learn a sharing parameter initialization for all tasks. The model shares structure across all tasks but is locally updated for task-specific predictions. Therefore, our framework can be decomposed into several sub-networks, where we only share the parameter initialization in the base learner where gradient boosting modules (the other weak learners) focus on making task-specific predictions.
> 3.	We have explained “step k” and gradient boost in Section 3. Specifically, the beginning of Section 3 gives the high-level idea of gradient boosting, and Section 3.1 gives detailed definitions. As each weak learner can be regarded as a neural network, theta-k represents the parameter of the k-th weak learner. We think it has been clearly stated in the paper.
>
> Regarding the **figure quality**, we will improve the figure quality in the next version.
>
> Regarding the **results**, we attribute the better results obtained by the proposed framework and related work than the plain MAML to that “providing more task-specific initialization tends to deliver better performance”.
>
> Regarding the **ability in dealing with multi-mode distributions**, the core idea of MAML is learning a sharing parameter initialization for all tasks. The model shares structure across all the tasks but is locally updated for task-specific predictions. The single initialization means that all the task models start from a single sharing condition. For the case that the tasks have multi-mode distributions, the traditional MAML simply delivers a 'mean' situation for all tasks. Under such a situation, tasks with “diverse” situations (as opposed to the 'mean' situation) will take more time in learning their task-specific model. A model with more flexibility in providing diverse (or task-specific) model initialization could alleviate this issue. Since our model aims to capture more task-specific knowledge, more gradient boost modules will weaken the contribution of the base learner to the final prediction. That is why we say the updating strategy ensures a “weak base learner”.

---

### Official Review · AnonReviewer3 · 2020-10-25
**This paper presents a gradient boosting framework for Meta-optimization, wherein an initialization condition can be generalized to different distributions related to different tasks. The main idea is learning a general base learner across all the distributions of different tasks, and later learning gradient boosted modules that are adjusted to each specific task. In this study, a deep neural net is used to build the ensemble of sub-networks.**

**Rating:** 6
**Confidence:** 3

**Review:**

This study is presented clearly, and the core idea is interesting. However, the presented novelty is limited to a globally (for all tasks) and locally (task-specific)  learning paradigm using a framework inspired by (Badirli et al., 2020). The authors have presented experimental results for both regression and classification setups, which are interesting.
In my opinion, the paper has relatively high quality and can be interesting for the ICLR community.
One question regarding Figure 4 (a), you have mentioned that adding more weak learners causes difficulties in capturing multi-mode patterns, but from this figure, one can see that it is not completely true for the early epochs (before 400), comparing the two weak learners versus one weak learner case and it seems more like fluctuations, how would you explain it.

---

> ### Author Response · Authors · 2020-11-22
> **To reviewer 3**
>
> Thank you for your comments. Regarding Figure 4 (a), the model’s performance exhibits more fluctuations when it is equipped with two or more weak learners as the base learner. This might be attributed to the updating strategy for the base learner and the following weak learners. For the case of two weak learners, the model’s performance fluctuates in the first few learning epochs but ramps up rapidly afterward, indicating the weak learners can capture more sharing knowledge across tasks, although the knowledge may be corrupted by the “dynamic” updating strategy. Some previous theoretical studies found that the gradient for the earlier weak learner decays with the increasing number of gradient boost modules. Our other training results for the model with two weak learners as the base learner also fluctuate in the first few epochs. We will do more experiments to analyze how the contribution of each part changes during the training process in the future version.

---

### Official Review · AnonReviewer1 · 2020-10-28
**An effective yet quite restricted approach**

**Rating:** 5
**Confidence:** 4

**Review:**

The paper proposed a method to incorporate neural networks into gradient boosting for meta-learning with a limited number of samples in each task.

The simulated experiments, where the generated tasks contain four continuous function, demonstrated that the proposed model outperformed MAML and its related variants, and on the classification task, it performed similarity to a previously proposed method MMAML.

In general, I am very interested in seeing neural networks being combined with boosting algorithms, but I do have quite a few questions.

1. The paper claimed that the learning and updating strategy proposed in the method ensured a weak base learner. I have to say that I am not satisfied by the claim. One can simply construct a linearly-separable dataset, which doesn't contribute to the overall tasks, and it only takes few updates for a two-layer neural network to converge, then the learner itself is not weak anymore. In addition, it adds noise to other tasks in the task set.

2. Normally, when a weak learner is learnt, a 1-D line search is conducted to find the optimal contribution of this particular learner. Although, shrinkage is usually applied in practice to anneal the contribution of new learners as a regularisation method to avoid overfitting. The paper seemed to have ignored the line-search part which can be done efficiently in 1D, whilst only the shrinkage was applied in the modelling, and I was wondering if there was a specific reason for that.

3. In terms of the application of the proposed algorithm, I think it is rather limited. My understanding is that, although the proposed algorithm worked better than MAML and its variants on simulated data, the algorithm at the same time involves many hyperparameters, including (1) the design of the base feature extractor, (2) the design of gradient boost modules, (3) the boosting rates, and the annealing of them, (4) number of local update steps,  (5) number of global update steps, and many other hyperparams regarding to the training of neural networks. The first two hyperparameters or designs are critical in the success of the gradient boosting algorithm as weak learners are needed for learning so that the gradient is informative for the subsequent learners. Tuning those hyperparameters could be a time-consuming and also non-trivial task itself already, compared to the MAML algorithm itself, or generally multi-task learning approaches for meta learning, this algorithm doesn't seem to be outstanding.

4. A fair comparison, IMO, could be to use the same feature extractor for producing vector representations of samples, and then directly apply gradient boosting with trees for meta learning. As mentioned in the paper, the authors also agreed that the feature extractors themselves have a huge impact on the final performance, therefore, it might be a good practice to check how well gradient boosting is able to handle meta-learning on top of extracted features.

---

> ### Author Response · Authors · 2020-11-22
> **To reviewer 1**
>
> Thanks for your careful reviews that help improve the quality of our work. We respond to your comments point by point below.
> 1. Regarding the weak learner, we believe the reviewer has indeed pointed out a very good question. Some theoretical studies have found that the gradient for the earlier weak learners decays with the increasing number of gradient boost modules. We did some experiments to evaluate the contribution of each part to the final prediction with different gradient boosting steps. We will put the results and some discussion in the future version.
> 2. Thanks for your suggestion on the regression tasks. We will consider line search methods in future works.
> 3. We tried the best to apply the same settings to the compared methods to make a fair comparison. Our framework takes more time for training because it contains more gradient boosting modules and requires more update steps. But our experimental results show that it does not take a large number of gradient boosting modules and update steps to obtain an acceptable result.
> 4. Several studies [1, 2] have shown that the choice for feature extractor can affect the final prediction results. We decided to choose a commonly used feature extractor, 4-CONV, to ensure a fair comparison. We tested the performance of several pre-trained models to obtain the image embedding (e.g. ResNet-12), and the results showed the pre-trained image embedding delivers better performance.
>
> [1] Sun et al. Meta-Transfer Learning for Few-Shot Learning, CVPR 2019.
>
> [2] Tian et al. Rethinking Few-Shot Image Classification: a Good Embedding Is All You Need? ECCV 2020.

---

### Official Review · AnonReviewer2 · 2020-10-28

**Rating:** 4
**Confidence:** 5

**Review:**

### Paper summary
This paper addresses a problem of model-agnostic meta-learning (MAML) and most of its variations/extensions - learning only a single parameter initialization for the entire task distribution, which might not be effective when the task distribution is too diverse. Inspired by gradient boosting, which aims to train a new learner to predict the residuals of the previously predicted result for each step, this paper proposes a meta gradient boosting framework. Specifically, the proposed framework represents the meta-learned global initialization as a base learner, consisting of the first few weak learners, which is responsible for acquiring sharable transferable knowledge by learning across all tasks (or task modes). Then, many gradient-boosting modules learn to capture task-specific information to fit diverse distributions more effectively. This paper evaluates the proposed framework and the baselines, including MAML, Multimodal MAML, and LEO, on both the few-shot regression task and the few-shot image classification task. To create diverse task distributions, it combines different function families (linear, sinusoidal, etc.) for the regression and merges multiple datasets (Omniglot, miniImageNet, etc.) for the classification. The experiments show that the proposed meta-gradient boosting framework (with 5 gradient boosting modules) achieves better or competitive results compared to the baselines. Ablation studies justify some design choices of the proposed framework, including the architecture of the weak learner, the number of gradient-boosting modules, the updating strategy for the gradient boosting modules, the boosting rate, etc. I believe this work studies an important problem and proposes an interesting framework. Yet, I have a few concerns regarding experimental setup and results which prevent me from accepting this paper (see below for details).

### Paper strengths
**motivation**
The motivation for addressing the issue of model-agnostic meta-learning (MAML) - learning only a single parameter initialization for the entire task distribution, which might not be effective when the task distribution is too diverse, is convincing.

**novelty**
As far as I am concerned, the idea of utilizing the intuition of gradient boosting is novel and interesting. This paper presents a reasonable way to implement this idea.

**technical contribution & ablation study**
Ablation studies provide insights that help understand the proposed framework and justify many design choices. This includes the architecture of the weak learner, the number of gradient boosting modules, the updating strategy for the gradient boosting modules, the boosting rate, etc.

**clarity**
The writing is very clear and the figures illustrate the ideas well. Also, the organization of the paper is easy to follow.

**experimental results**
- The description of the experimental setup is comprehensive. The presentation of the experimental results is clear.
- The proposed meta-gradient boosting framework outperforms or at least performs competitively compared to the representative baselines.
- This paper studies a variety (i.e. different numbers of task modes) of settings, which provides great insights.

### Paper weaknesses
**baselines**
I believe this paper ignores many important baselines. As a result, the experimental conclusions are less convincing. I list some of the baselines and brief reasons why I believe they should be included as below:
- Hierarchically Structured Meta-Learning (HSML): is designed to perform soft clustering on tasks. It would be interesting to see if HSML can handle the task distributions considered in this paper. Also, it shows superior performance compared to MAML and a workshop version of Multimodal MAML. So, it is a stronger baseline to be included.
- Probalistic MAML / Bayesian MAML: both of these two methods consider , outperforming MAML. Intuitively, this probabilistic schema should deal with multimodal task distributions better since it inherently learns to handle uncertainty. It would be great to compare against at least one of them.
- Proto-MAML: presented in the meta-dataset paper (Triantafillou et al. in ICLR 2020), is designed to deal with multiple datasets combined and therefore should be able to perform well on the setup considered in this paper. Proto-MAML shows the strongest performance compared to MAML based methods and even outperforms some metric-based meta-learning methods. Showing the proposed framework can outperform or perform competitively compared to Proto-MAML would make this paper much stronger.
- Metric-based meta-learning methods: this paper does not include comparisons against any metric-based meta-learning methods such as matching networks, prototypical networks, relation networks, TADAM, etc. Yet, the state-of-the-art results on the few-shot image classification are mostly achieved by metric-based meta-learning methods. Therefore, I believe it would be essential to include representative metric-based meta-learning methods.

**MAML with a comparable number of parameters**
Since the proposed meta-gradient boosting framework has more parameters than the vanilla MAML, it is possible that the performance gain comes from the larger capacity. It would important to include a MAML baseline that has a comparable number of parameters to justify this.

**RL experiments**
As far as I am concerned, the few-shot regression task is more a task for detailed analysis for research rather than a task with a wide range of applications, and the state-of-the-art results of the few-shot image classification task have been mostly achieved by metric-based meta-learning methods. Therefore, I am mainly interested in the model-agnostic meta-learning  line of work because of its potential in reinforcement learning where the ability to adapt to unseen scenarios is crucial. Yet, this paper does not include any experiments on RL without any reasons, which makes the paper less convincing to me.

**MGB-1 vs. MAML**
It seems that the proposed meta-gradient boosting model with one gradient-boosting module outperforms the vanilla MAML by a significant margin on the regression task yet only performs similarly to the vanilla MAML on the classification. Can the authors give some intuition about why this is the case?

**meta-dataset**
To evaluate if the proposed framework and the baselines can deal with diverse task distributions on the few-shot image classification task, this paper combines four different few-shot learning datasets to produce a 4-mode classification task. Yet, the meta-dataset has been proposed for this purpose. Also, the meta-dataset paper provides a comprehensive comparison of recent meta-learning methods. Therefore, I believe this paper should evaluate the proposed framework on the meta-dataset and see if it outperforms the baselines.

**stddev of the classification task**
It seems that the performance gap between the baselines and the proposed framework is insignificant on the image classification task. In this case, it would be important to also provide the standard deviation of each task.

=== After rebuttal ===

I am not satisfied with the response from the authors. I can only hardly recognize the effort made by the authors during the rebuttal - most of my points were only briefly discussed in the response without revising the paper.

- The suggested baselines were merely briefly discussed but not added to the comparison.
- The results of the meta-dataset, which in my opinion is the most suitable dataset for the purpose, are still not included in the revised paper.
- The stddev of the classification task is not still not provided, making it hard to justify the performance gain.
- Why is RL left to future work?

I have read the reviews from other reviewers. With the little revision from the authors, I have decided to keep my original rating and would not recommend this paper to be accepted.

---

> ### Author Response · Authors · 2020-11-22
> **To reviewer 2**
>
> Thank you for the detailed reviews and suggestions.
>
> **Baseline.** Regarding the choice of baselines, we believe some of the baselines suggested by the reviewer are unsuitable for our case. We present the reasons below:
> 1. Regarding HSML, we did investigate HSML in our related work (in the 2nd paragraph of Related Work) but chose MMAML as a baseline instead of HSML, in light of that they share a similar spirit. We appreciate that you kindly suggest that “HSML shows superior performance compared to MAML and a workshop version of MMAML”. We will definitely look into the performance of HSML and add more experiments in the future version.
> 2. Regarding probabilistic MAML, the compared method, LEO, is, actually, a probabilistic MAML and should suffice to serve as a representative of methods in the same category. Specially, LEO considers the multiple conditions by borrowing the idea of variational autoencoders and uses an encoder to learn the mean and variance for generating a weight matrix for the predictor.
> 3. Proto-MAML is a metric-based approach for image classification. For reasons why we did not compare our work with metric-based methods, please see below (our response #4).
> 4. We did not choose metric-based methods for comparison for two reasons. First, we aim to provide a general framework for solving a wide variety of tasks in this work. However, metric-based methods usually calculate the relationship between new samples and history samples in terms of a relation score or a probabilistic distribution (for classification), which is unsuitable for few-shot regression tasks because it is hard to learn from very few points to generate a wide range for the outputs. Second, our focus is the single initialization problem addressed by traditional MAML-based frameworks. Therefore, MAML-based frameworks naturally become the major baselines to compare. And we think the reason why ‘metric-based methods superior other methods in few-shot image classification tasks’ will be another interesting research topic, due to the learning strategy of the metric-based approaches.
>
> **meta-dataset** About the meta-dataset. In this work, we followed the previous settings (e.g., those in MMAML) to set up the multi-mode tasks. But we reckon that the meta-dataset could be a valuable source and would definitely consider it as a good option for few-shot image classification tasks.
>
> **MGB-1 vs. MAML** we also noticed the difference between MGB-1 vs. MAML in their regression and classification results. We have explained in the paper that “MGB-1 can make only a slight improvement over MAML because images contain more complex information than real numbers” for regression tasks. The MGB-1 with one gradient boost module can capture more information than MAML. The reason may be related to the network dimension and structure. So we also appreciate your comments about ‘MAML with a comparable number of parameters’. We will add some experiments about “MAML with a comparable number of parameters” in the future version.
>
> **MAML with a comparable number of parameters** We have set the same feature extractor for all methods in our current experimental settings to ensure a fair comparison. In further experiments, we will explore whether the improvement comes from the gradient boosting module or the enlarged network structure.
>
> **stddev of the classification task**: we did not present stddev for the classification task due to the limited space. We will add more experimental details regarding this in the future version.
>
> **RL experiments**: We will leave experiments on reinforcement learning to our future work.

---

### Decision · Program_Chairs · 2021-01-07
**Final Decision**

**Decision:**

Reject

**Comment:**

The paper proposes a meta-gradient boosting framework to tackle the model-agnostic meta-learning problem. The idea is to use a base learn that learns shared information across tasks, and gradient boosted modules to capture task-specific modules. The experiments show that the proposed meta-gradient boosting framework (with 5 gradient boosting modules) achieves better or competitive results compared to the baselines. However, there were several issues that the author feedback did not addressed properly. For instance, R2 were not satisfied by discussing briefly the suggested baselines without adding the comparison, or R1 pointed out that the claim “the learning and updating strategy proposed in the method ensured a weak base learner” because clear separable datasets could convergence quickly and weak is not anymore applicable. Besides these two specific concerns, the reviewers expected a large revision of the paper due to several cons about the paper. All reviewers agreed a mayor revision is needed before acceptance. Therefore I recommend rejection.